# The prevalence of metabolic conditions before and during the COVID-19 pandemic and its association with health and sociodemographic factors

Hadii M. Mamudu[1,2]*, David Adzrago[3], Emmanuel O. Odame[4], Oluwabunmi Dada[5], Valentine Nriagu[1,2], Trishita Paul[6], Florence W. Weierbach[1,7], Karilynn Dowling-McClay[1,8], David W. Stewart[1,8], Jessica Adams[9], Timir K. Paul[1,10]

1 The Center for Cardiovascular Risk Research, College of Public Health, East Tennessee State University, Johnson City, TN, United States of America, 2 Department of Health Services Management and Policy, College of Public Health, East Tennessee State University, Johnson City, TN, United States of America, 3 Center for Health Promotion and Prevention Research, CDC Prevention Research Center, School of Public Health, University of Texas Health Science Center, Houston, TX, United States of America, 4 Department of Environmental Health Sciences, The University of Alabama at Birmingham, Birmingham, AL, United States of America, 5 Department of Occupational Safety and Health, Murray State University, Murray, KY, United States of America, 6 Brentwood High School, Brentwood, TN, United States of America, 7 College of Nursing, East Tennessee State University, Johnson City, TN, United States of America, 8 College of Pharmacy, East Tennessee State University, Johnson City, TN, United States of America, 9 Lone Star College of Nursing, Lone Star College–North Harris, Lone Star College, Houston, TX, United States of America, 10 The University of Tennessee at Nashville, Ascension St. Thomas Hospital, Nashville, TN, United States of America

* mamudu@etsu.edu

## Abstract

### Background

There is a dearth of evidence on the relationship between COVID-19 and metabolic conditions among the general U.S. population. We examined the prevalence and association of metabolic conditions with health and sociodemographic factors before and during the COVID-19 pandemic.

### Methods

Data were drawn from the 2019 (N = 5,359) and 2020 (N = 3,830) Health Information National Trends Surveys on adults to compare observations before (2019) and during (2020) the COVID-19 pandemic. We conducted weighted descriptive and multivariable logistic regression analyses to assess the study objective.

### Results

During the pandemic, compared to pre-pandemic, the prevalence of diabetes (18.10% vs. 17.28%) has increased, while the prevalence of hypertension (36.38% vs. 36.36%) and obesity (34.68% vs. 34.18%) has remained similar. In general, the prevalence of metabolic conditions was higher during the pandemic (56.09%) compared to pre-pandemic (54.96%).

**Funding:** The author(s) received no specific funding for this work.

**Competing interests:** The authors have declared that no competing interests exist.

Compared to never smokers, former smokers had higher odds of metabolic conditions (AOR = 1.38, 95% CI = 1.01, 1.87 and AOR = 1.57, 95% CI = 1.10, 2.25) before and during the pandemic, respectively. People with mild anxiety/depression symptoms (before: AOR = 1.52, 95% CI = 1.06, 2.19 and during: AOR = 1.55, 95% CI = 1.01, 2.38) had higher odds of metabolic conditions relative to those with no anxiety/depression symptoms.

## Conclusion

This study found increased odds of metabolic conditions among certain subgroups of US adults during the pandemic. We recommend further studies and proper allocation of public health resources to address these conditions.

## Introduction

The SARS-CoV-2 (Coronavirus diseases; COVID-19) has continued to affect many countries, including the United States, since the World Health Organization (WHO) declared a global pandemic in March 2020 [1, 2]. Before the declaration, metabolic conditions such as obesity, type 2 diabetes mellitus, and cardiovascular disease have continued to be the leading cause of morbidity and mortality in the U.S. and the world [3–5]. In 2018, approximately 13% of all U.S. adults had diabetes, with 2.8% of this population being unaware of their status but meeting laboratory criteria for diabetes [6]. Similarly, a national survey from 2017 to 2018 shows that 42.4% of U.S. adults had obesity [7]. These metabolic conditions are associated with severe health risks [8, 9]. Additionally, they predispose people to the risks of death and adverse health outcomes from COVID-19 [8–13]. However, studies on the effects of COVID-19 on these metabolic conditions are scarce. Indeed, the associations between cardiometabolic conditions in U.S. adults and the COVID-19 pandemic and risk factors such as physical inactivity, tobacco use, anxiety/depression, and sociodemographic characteristics remain understudied.

Not only did patients with diabetes or obesity have increased mortality due to COVID-19 infection [14–20], their overall health was also negatively impacted by the COVID-19 lockdown [17, 19]. Studies demonstrated poor glycemic control and increased body mass index (BMI) for patients with diabetes during the lockdown [20, 21] along with a deterioration in glucose regulation [16, 22]. The timings of lockdown orders in the target populations from these studies differ, as they were based on when the countries declared a lockdown [16, 20–22]. Other studies have shown that newly diagnosed diabetes is more prevalent in patients following COVID-19 infection, with one in every ten COVID-19 patients diagnosed with new onset diabetes mellitus [15, 23]. In addition to metabolic health outcomes secondary to diabetes, some subgroups, specifically those with cardiometabolic disease [24], were at an increased risk of poor mental health outcomes, had depressive symptoms [25], and poor sleep quality [26]. Other negative behavioral activities included reduced physical activity [27, 28] and increased alcohol consumption [26].

While the metabolic conditions such as diabetes and obesity are well documented, there is a paucity of research on the association between conditions such as hypertension and COVID-19 [29]. Given the high prevalence of medical conditions such as hypertension and the potential for negative health outcomes secondary to COVID-19, this study explores the relationships between COVID-19 and metabolic conditions before and during the COVID-19 pandemic.

We utilize a nationally representative sample of U.S. adults to estimate the prevalence of metabolic conditions (diabetes, hypertension, and obesity) before and during the COVID-19 pandemic declaration. Further, this study examines the association between these metabolic conditions among U.S. adults and the ramifications of the COVID-19 pandemic, including

physical inactivity, tobacco use, anxiety/depression, and sociodemographic characteristics. Considering the higher prevalence of these metabolic conditions in the U.S. adult population and throughout the world, it is imperative to understand which populations to target for public health interventions to decrease COVID-19 related morbidities for high-risk populations.

## Methods

The 2019 and 2020 Health Information National Trends Surveys (HINTSs) de-identified public-use datasets were combined for this study. HINTS is a cross-sectional survey that assesses health-related information (e.g., diabetes, hypertension, and obesity) and behaviors (e.g., tobacco use) among a nationally representative sample of U.S. adults aged ≥18 years. It uses a random sampling technique to select a sample of the U.S. civilian, noninstitutionalized adult population. Details of the methods, questionnaire, and survey administration have been published [30, 31]. The 2019 survey (HINTS 5 Cycle 3) was conducted from January through April 2019, and the 2020 survey (HINTS 5 Cycle 4) was conducted from February through June 2020. These surveys are the recent publicly available HINTS datasets. The combined HINTS 5 Cycles 4 (N = 3,865) and 3 (N = 5,438) datasets consist of a total sample of 9,303 adults. The study was reviewed by the Institutional Review Board (IRB) of East Tennessee State University and exempted as the HINTS datasets are de-identified and publicly available.

## Measures and variables

### Dependent variable

The dependent variable is "metabolic condition", derived from three distinct questions about diabetes, hypertension, and obesity. For diabetes, the participants were asked, "Has a doctor or other health professional ever told you that you have diabetes or high blood sugar?" (yes/no). Hypertension was assessed with the question, "Has a doctor or other health professional ever told you that you have high blood pressure or hypertension?" (yes/no). Obesity status was determined using body mass index (BMI), and defined as underweight = <18.5, healthy/normal = 18.5–24.9, overweight = 25.0–29.9, and obese ≥ 30.0. Obese was defined as BMI ≥ 30.0 and not obese as BMI < 30 [32, 33]. Thus, the variable "metabolic condition" in this study was ascertained as participants who had diabetes, hypertension, or were obese.

### Independent variables

The main independent variable is the HINTS survey year, which was based on the 2019 and 2020 surveys, given that COVID-19 cases were widespread globally by January 2020 [31, 34, 35]. The 2019 HINTS data were used as the pre-COVID-19 pandemic cohort, while the 2020 HINTS data were used as the COVID-19 pandemic cohort for stratified analysis.

Other independent variables analyzed in this study included self-reported sociodemographic characteristics, moderate physical activity intensity, cigarette smoking status, e-cigarette use status, and anxiety/depression symptoms [36–38]. The sociodemographic variables included age (18–25, 26–34, 35–49, 50–64, and 65+), sex (male/female), race/ethnicity (non-Hispanic White, non-Hispanic Black/African American, Hispanic, non-Hispanic Asian, and non-Hispanic others), gender identity (heterosexual/straight or sexual minorities [homosexual, lesbian, gay, or bisexual]), marital status (single/never married, married/living as married, divorced/separated, or widowed), level of education completed (less than high school, high school graduate, some college, and college graduate or higher), and total family income (<$20,000, $20,000 to < $35,000, $35,000 to < $50,000, $50,000 to < $75,000, or ≥ $75,000). General health status was based on self-ratings of overall health as excellent, very good, good,

fair, or poor. Due to limited samples, we dichotomized general health status into excellent/very good/good or fair/poor. The number of days per week of moderate intensity physical activity (none and at least one day per week), cigarette smoking status (never/non-smoker, former smoker, and current daily or some days smoker), and e-cigarette use status (never used, former user, and current daily/some days user) were also included.

The anxiety/depression symptoms variable was constructed from Patient Health Questionnaire-4 (PHQ-4) in the HINTS 5 survey. The PHQ-4 assesses symptoms/signs of anxiety and depression, with total scores from 0–12 (0–2 = normal/negative, 3–5 = mild, 6–8 = moderate, and 9–12 = severe) [39, 40]. Thus, anxiety/depression symptoms were categorized into normal or no anxiety/depression, mild, moderate, and severe.

## Statistical analyses

The HINTS sampling weight was applied to the analysis to achieve population estimates and offset nonresponse. We estimated the weighted prevalence of each component of metabolic condition before and during the COVID-19 pandemic. The weighted prevalence and unweighted frequencies of metabolic conditions by the sociodemographic characteristics, moderate-intensity physical activity, cigarette smoking status, e-cigarette use status, and anxiety/depression symptoms were computed to characterize the survey sample before and during the COVID-19 pandemic. Additionally, two logistic regression analyses represented by two models were conducted. While Model 1 assessed the association between the metabolic conditions and independent variables using the data before the pandemic, Model 2 utilized the data during the pandemic. All analyses were weighted using the HINTS sampling weight and replicate weight to offset non-response bias and to achieve nationally representative estimates [30, 31]. The weighted percentages, adjusted odds ratios (AOR), 95% 2-tailed confidence intervals (CI), and statistically significant $p$-value ($< 0.05$) have been reported.

## Results

The prevalence of each metabolic condition (diabetes, hypertension, and obesity) before and during the COVID-19 pandemic is presented in Fig 1. The results showed that the prevalence of diabetes was higher during the COVID-19 pandemic (18.10%) than before the pandemic (17.28%). However, the prevalence of hypertension (36.36% vs. 36.38%) and obesity (34.68% vs. 34.18%) was similar during and before the pandemic.

Table 1 presents the prevalence of metabolic conditions by independent variables before and during the COVID-19 pandemic. Overall, the prevalence of metabolic conditions (54.96% vs. 56.09%) was higher during the pandemic than before the pandemic. The distribution of the prevalence of metabolic conditions within the sociodemographic groups, moderate intensity physical activity, cigarette smoking status, and e-cigarette use status before and during the COVID-19 pandemic varied. The prevalence of metabolic conditions during the COVID-19 pandemic, compared to before the pandemic increased for individuals aged 35–49 and 50–64 years but decreased for those aged 18–25, 26–34, and ≥65 years. The prevalence also increased for all non-Hispanic racial/ethnic groups but decreased for Hispanic individuals. The prevalence of metabolic condition was higher for individuals who did not engage in moderate-intensity physical activity compared to those who engaged in at least one moderate-intensity physical activity per week. Individuals who were former cigarette smokers or current smokers had an increased prevalence of metabolic conditions. For e-cigarette use groups, the prevalence had increased for those who had never used e-cigarettes and those who currently used e-cigarettes; however, it decreased for former e-cigarette users.

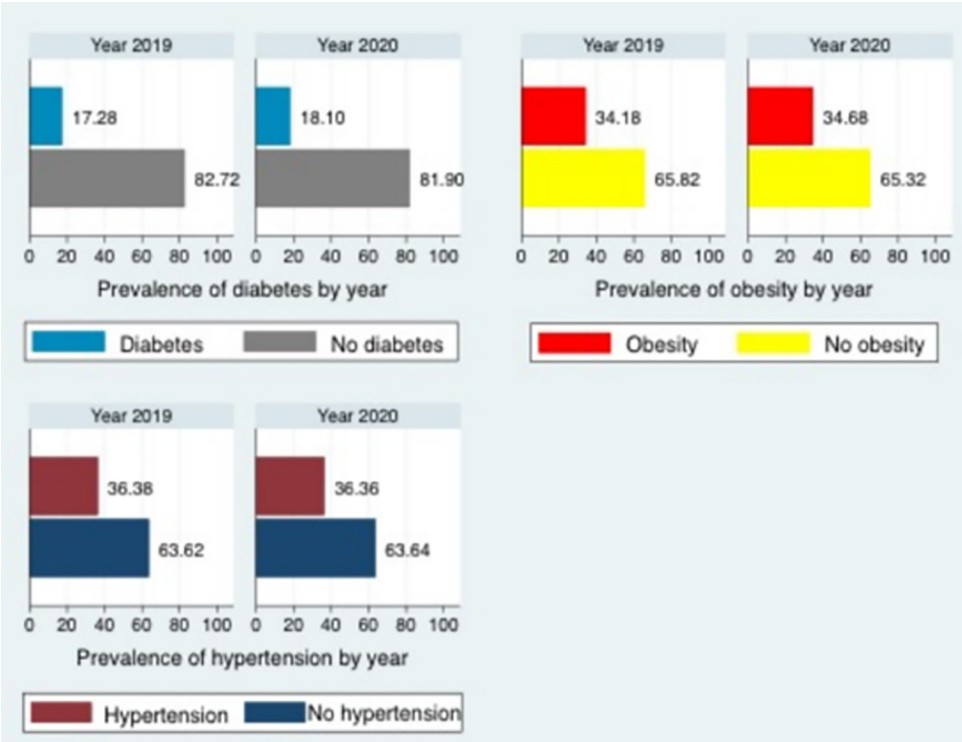

**Fig 1. The prevalence of diabetes, obesity, and hypertension before (2019) and during (2020) the COVID-19 pandemic.**

Table 2 shows metabolic conditions and their associated factors before (Model 1) and during (Model 2) the COVID-19 pandemic, respectively. Before the pandemic, compared to age 18–25 years, only two groups of individuals aged 50–64 (AOR = 2.64, 95% CI = 1.20, 5.77) and 65 years or older (AOR = 4.82, 95% CI = 2.25, 10.32) had significantly higher odds of metabolic conditions. During the pandemic, the likelihood of metabolic conditions was significantly higher for four groups: individuals aged 26–34 (AOR = 1.95, 95% CI = 1.04, 3.67), 35–49 (AOR = 4.13, 95% CI = 2.12, 8.04), 50–64 (AOR = 6.19, 95% CI = 3.03, 12.65), and 65 years or older (AOR = 7.82, 95% CI = 3.92, 15.57) compared to age 18–25 years. During the pandemic, males had significantly higher odds of metabolic conditions (AOR = 1.28, 95% CI = 1.01, 1.64) relative to females, whereas the odds were not different before the pandemic. Compared to non-Hispanic White people, the odds were significantly higher for non-Hispanic Black people before (AOR = 2.01, 95% CI = 1.26, 3.22) and during (AOR = 2.09, 95% CI = 1.22, 3.58) the pandemic. Engaging in at least one moderate-intensity physical activity per week was associated with a lower likelihood of metabolic conditions before (AOR = 0.64, 95% CI = 0.46, 0.88) and during (AOR = 0.58, 95% CI = 0.42, 0.79) the pandemic as compared to no physical activity. Having mild (AOR = 1.52, 95% CI = 1.06, 2.19) or severe (AOR = 2.44, 95% CI = 1.27, 4.69) anxiety/depression symptoms, compared to no anxiety/depression symptoms, was associated with higher metabolic conditions before the pandemic. Only mild anxiety/depression symptoms (AOR = 1.55, 95% CI = 1.01, 2.38) were associated with higher metabolic conditions during the pandemic. Compared to people who never smoked cigarettes, former cigarette smokers had significantly higher odds of metabolic conditions before (AOR = 1.38, 95% CI = 1.01, 1.87) and during (AOR = 1.57, 95% CI = 1.10, 2.25) the pandemic, but not current smokers. Before the pandemic, the likelihood of metabolic conditions was significantly lower

**Table 1. Demographic characteristics of U.S. adults before and during the COVID-19 pandemic and association with metabolic conditions (N = 9,189).**

| | Before the COVID-19 pandemic (2019) | | | During the COVID-19 pandemic (2020) | | |
|---|---|---|---|---|---|---|
| | Metabolic conditions | | | Metabolic conditions | | |
| | Total, n | n(%) | | Total, n | n(%) | |
| Characteristics | (n = 5,359) | 3,287 (54.96) | *P* value | (n = 3,830) | 2,353 (56.09) | *P* value |
| **Age** | | | < .001 | | | < .001 |
| 18–25 | 187 (11.78) | 51 (32.54) | | 147(13.32) | 44 (29.31) | |
| 26–34 | 493 (12.49) | 170 (38.38) | | 337 (12.97) | 118 (37.98) | |
| 35–49 | 964 (24.61) | 444 (47.75) | | 701 (25.47) | 342 (55.97) | |
| 50–64 | 1,654 (31.11) | 1,067 (61.25) | | 1,137 (27.78) | 746 (65.53) | |
| 65+ | 1,930 (20.02) | 1,470 (76.64) | | 1,396 (20.45) | 1,042 (73.34) | |
| **Sex** | | | .577 | | | .288 |
| Female | 2,795 (50.77) | 1,677 (54.53) | | 2,038 (50.88) | 1,242 (55.07) | |
| Male | 2,091 (49.23) | 1,320 (55.95) | | 1,486 (49.12) | 934 (57.75) | |
| **Race/ethnicity** | | | .002 | | | < .001 |
| Non-Hispanic White | 3,030 (63.49) | 1,740 (53.04) | | 2,125 (63.33) | 1,245 (55.55) | |
| Non-Hispanic Black | 669 (11.26) | 505 (64.76) | | 478 (11.145) | 369 (70.58) | |
| Hispanic | 724 (16.83) | 444 (59.00) | | 595 (16.96) | 343 (49.20) | |
| Non-Hispanic Asian | 223 (5.35) | 111 (37.57) | | 160 (5.23) | 81 (44.75) | |
| Non-Hispanic other | 165 (3.07) | 100 (48.46) | | 119 (3.34) | 72 (67.55) | |
| **Gender identity** | | | .610 | | | .658 |
| Heterosexual | 4,747 (95.65) | 2,896 (54.96) | | 3,389 (94.57) | 2,083 (56.45) | |
| Homosexual or lesbian/gay or bisexual | 194 (4.35) | 113 (51.12) | | 163 (5.43) | 92 (53.50) | |
| **Marital status** | | | < .001 | | | < .001 |
| Single/never married | 876 (30.52) | 468 (45.59) | | 644 (30.87) | 354 (49.31) | |
| Married/living as married | 2,827 (55.69) | 1,654 (56.43) | | 1,972 (54.77) | 1,177 (58.39) | |
| Divorced/separated | 936 (8.94) | 623 (63.57) | | 679 (9.75) | 445 (58.45) | |
| Widowed | 581 (4.85) | 461 (78.85) | | 410 (4.61) | 310 (77.46) | |
| **Level of education completed** | | | < .001 | | | < .001 |
| Less than High School | 327 (6.88) | 239 (65.38) | | 273 (8.05) | 202 (66.33) | |
| High School graduate | 932 (23.36) | 675 (63.18) | | 702 (22.47) | 509 (66.69) | |
| Some college | 1,580 (40.26) | 1,052 (57.32) | | 1,075 (39.16) | 707 (56.87) | |
| College graduate or higher | 2,394 (29.50) | 1,243 (43.09) | | 1,658 (30.32) | 865 (45.42) | |
| **Total annual family income** | | | < .001 | | | < .001 |
| <$20,000 | 883 (18.35) | 636 (62.99) | | 619 (15.09) | 455 (67.22) | |
| $20,000 - $34,999 | 608 (11.05) | 417 (62.90) | | 450 (11.48) | 298 (61.18) | |
| $35,000 - $49,999 | 623 (13.48) | 412 (59.33) | | 459 (12.70) | 300 (62.05) | |
| $50,000 - $74,999 | 845 (17.46) | 524 (55.54) | | 589 (18.20) | 351 (54.86) | |
| ≥$75,000 | 1,795 (39.66) | 915 (46.42) | | 1,319 (42.54) | 695 (48.49) | |
| **General health status** | | | < .001 | | | < .001 |
| Excellent/very good/good | 4,478 (84.84) | 2,555 (50.08) | | 3,187 (85.89) | 1,818 (51.72) | |
| Fair/poor | 853 (15.16) | 713 (81.98) | | 626 (14.11) | 525 (82.99) | |
| **Moderate physical activity intensity** | | | < .001 | | | < .001 |
| None | 1,421 (25.78) | 1,065 (68.23) | | 1,041 (27.12) | 773 (69.39) | |
| At least one day per week | 3,861 (74.22) | 2,169 (50.26) | | 2,739 (72.88) | 1,545 (51.01) | |
| **Anxiety/depression symptoms** | | | .001 | | | .229 |
| None | 3,771 (68.35) | 2,252 (52.23) | | 2,669 (68.57) | 1,594 (53.65) | |
| Mild | 865 (18.08) | 533 (55.43) | | 629 (17.48) | 398 (59.62) | |
| Moderate | 334 (7.40) | 225 (57.83) | | 258 (7.93) | 174 (61.98) | |

*(Continued)*

**Table 1.** (Continued)

| | Before the COVID-19 pandemic (2019) | | | During the COVID-19 pandemic (2020) | | |
|---|---|---|---|---|---|---|
| | **Metabolic conditions** | | | **Metabolic conditions** | | |
| | **Total, n** | **n(%)** | | **Total, n** | **n(%)** | |
| Severe | 236 (6.18) | 174 (74.62) | | 173 (6.02) | 113 (58.89) | |
| **Cigarette smoking status** | | | < .001 | | | < .001 |
| Never | 3,261 (64.22) | 1,892 (51.74) | | 2,407 (63.11) | 1,403 (50.98) | |
| Former smoker | 1,406 (23.27) | 984 (63.86) | | 931 (23.02) | 628 (67.91) | |
| Current smoker | 611 (12.52) | 364 (55.43) | | 436 (13.87) | 287 (59.39) | |
| **E-cigarette use status** | | | < .001 | | | .005 |
| Never | 4,591 (80.77) | 2,882 (57.15) | | 3,297 (80.91) | 2,070 (58.40) | |
| Former user | 526 (13.83) | 282 (53.61) | | 381 (12.71) | 205 (52.51) | |
| Current user | 173 (5.40) | 86 (28.66) | | 114 (6.39) | 55 (37.24) | |

Data source: 2019 and 2020 Health Information National Trends Surveys, HINTS 5 Cycles 3 and 4, respectively.

Unweighted N = 9,189 and weighted N = 501,680,570.

Before the COVID-19 pandemic data (HINTS 5 Cycles 3) were collected from January through April 2019, and during the COVID-19 pandemic data were collected from February through June 2020.

Frequencies were not weighted, while percentages were weighted. Differences in total numbers in categories may be due to missing data.

for current e-cigarette users (AOR = 0.44, 95% CI = 0.23, 0.85) compared to those who had never used e-cigarettes, with no difference observed during the pandemic.

## Discussion

This study assessed the prevalence of metabolic conditions among U.S. adults and the underlying associated factors before and during the COVID-19 pandemic using the HINTS 2019 and 2020 survey data. To the best of our knowledge, this is the first study to use nationally representative U.S. adult data to highlight associations between metabolic outcomes, sociodemographic factors, and the COVID-19 pandemic.

There was an increase in the overall prevalence of metabolic conditions, especially among certain subgroups during the COVID-19 pandemic. This is consistent with a systematic review that assessed the impact of disasters, including pandemics, on metabolic conditions and reported increased incidence and mortality for diabetes and obesity [41]. Our findings indicate that being elderly (aged 50+), non-Hispanic Black person, former smoker, having fair/poor health status, and having mild anxiety significantly increased the likelihood of metabolic conditions pre- and during the pandemic. However, the disparities in these health and sociodemographic factors were greater during the pandemic.

Previous studies have established that the COVID-19 pandemic exacerbated and further unmasked existing disparities in metabolic outcomes [42–46]. For instance, we found that the odds of metabolic outcomes were significantly higher only among the elderly age groups (50–64 and 65+) compared to young adults before the pandemic. However, these increases in odds almost doubled among these age groups during the pandemic. Significantly higher odds were also noted in the middle age groups (26–34 and 35–49), where the odds almost tripled for the 35–49 age group during the pandemic. Consistent with the literature, our results indicated age was the strongest factor associated with an increased likelihood of adverse metabolic conditions during the pandemic [46–48] with higher age range conferring a higher risk for metabolic conditions. The association between age and metabolic conditions during the pandemic and the risks for adverse health conditions from COVID-19 suggests that health interventions

**Table 2. The odds of metabolic conditions among U.S. adults before and during the COVID-19 pandemic (N = 9,189).**

| | Before the COVID-19 (Model 1) | | During the COVID-19 (Model 2) | |
|---|---|---|---|---|
| | AOR | 95% CI | AOR | 95% CI |
| **Age** | | | | |
| 18–25 | Ref. | - | Ref. | - |
| 26–34 | 1.04 | (0.44, 2.45) | **1.95*** | **(1.04, 3.67)** |
| 35–49 | 1.46 | (0.66, 3.26) | **4.13****** | **(2.12, 8.04)** |
| 50–64 | **2.64*** | **(1.20, 5.77)** | **6.19****** | **(3.03, 12.65)** |
| 65+ | **4.82****** | **(2.25, 10.32)** | **7.82****** | **(3.92, 15.57)** |
| **Sex** | | | | |
| Female | Ref. | - | Ref. | - |
| Male | 1.11 | (0.85, 1.44) | **1.28*** | **(1.01, 1.64)** |
| **Race/ethnicity** | | | | |
| Non-Hispanic White | Ref. | - | Ref. | - |
| Non-Hispanic Black | **2.01**** | **(1.26, 3.22)** | **2.09**** | **(1.22, 3.58)** |
| Hispanic | 1.33 | (0.93, 1.89) | 0.91 | (0.64, 1.28) |
| Non-Asian | 0.74 | (0.44, 1.25 | 0.72 | (0.42, 1.25) |
| Non-Hispanic other | 1.37 | (0.56, 3.35) | 1.25 | (0.57, 2.72) |
| **Gender identity** | | | | |
| Heterosexual | Ref. | - | Ref. | - |
| Homosexual/lesbian/gay/bisexual | 1.43 | (0.67, 3.06) | 1.30 | (0.58, 2.93) |
| **Marital status** | | | | |
| Single/never married | Ref. | - | Ref. | |
| Married/living as married | **1.68**** | **(1.18, 2.39)** | 0.74 | (0.49, 1.13) |
| Divorced/separated | 1.40 | (0.92, 2.14) | **0.46**** | **(0.27, 0.78)** |
| Widowed | 1.89 | (0.97, 3.66) | 0.71 | (0.36, 1.40) |
| **Level of education completed** | | | | |
| Less than High School | Ref. | - | Ref. | - |
| High school graduate | 1.40 | (0.72, 2.71) | 1.23 | (0.611, 2.49) |
| Some college | 1.31 | (0.67, 2.55) | 0.93 | (0.45, 1.92) |
| College graduate/higher | 1.07 | (0.54, 2.14) | 0.72 | (0.35, 1.47) |
| **Total family annual income** | | | | |
| <$20,000 | Ref. | | Ref. | - |
| $20,000 - $34,999 | 1.02 | (0.61, 1.68) | **0.77** | **(0.38, 1.56)** |
| $35,000 - $49,999 | 0.94 | (0.54, 1.65) | 0.78 | (0.42, 1.45) |
| $50,000 - $74,999 | 0.89 | (0.52, 1.53) | 0.67 | (0.39, 1.17) |
| ≥$75,000 | 0.73 | (0.45, 1.17) | 0.63 | (0.37, 1.07) |
| **General health status** | | | | |
| Excellent/very good/good | Ref | - | Ref | - |
| Fair/poor | **2.82****** | **(1.99, 3.99)** | **2.99****** | **(1.97, 4.53)** |
| **Moderate physical activity intensity** | | | | |
| None | Ref | | Ref | |
| At least one day per week | **0.64**** | **(0.46, 0.88)** | **0.58****** | **(0.42, 0.79)** |
| **Anxiety/depression symptoms** | | | | |
| None | Ref. | | Ref. | - |
| Mild | **1.52*** | **(1.06, 2.19)** | **1.55*** | **(1.01, 2.38)** |
| Moderate | 1.31 | (0.83, 2.08) | 1.76 | (0.87, 3.57) |
| Severe | **2.44**** | **(1.27, 4.69)** | 1.53 | (0.72, 3.22) |
| **Cigarette smoking status** | | | | |

*(Continued)*

**Table 2.** (Continued)

|  | Before the COVID-19 (Model 1) | | During the COVID-19 (Model 2) | |
|---|---|---|---|---|
|  | AOR | 95% CI | AOR | 95% CI |
| Never smoker | Ref. |  | Ref. | - |
| Former smoker | **1.38**\* | **(1.01, 1.87)** | **1.57**\*\* | **(1.10, 2.25)** |
| Current smoker | 0.93 | (0.59, 1.46) | 1.02 | (0.54, 1.95) |
| **E-cigarette smoking status** |  |  |  |  |
| Never user | Ref. |  | Ref. | - |
| Former user | 1.03 | (0.63, 2.68) | 1.03 | (0.60, 1.78) |
| Current user | **0.44**\* | **(0.23, 0.85)** | 0.63 | (0.30, 1.30) |

Data source: 2019 and 2020 Health Information National Trends Surveys, HINTS 5 Cycles 3 and 4 respectively.

Before the COVID-19 pandemic data (HINTS 5 Cycles 3) were collected from January through April 2019, and during the COVID-19 pandemic data were collected from February through June 2020.

AOR = Adjusted odds ratio. 95% CI = 95% confidence interval. Ref = Reference group.

\*$p \leq 0.05$

\*\*$p \leq 0.01$

\*\*\*$p \leq 0.001$.

targeted at high-risk groups such as the elderly could optimize outcomes, particularly during disasters such as this global pandemic.

Moreover, this study provides additional evidence that individual health behaviors played a critical role in developing metabolic conditions before and during the pandemic. For example, while being a former smoker increased the odds of metabolic conditions, engaging in moderate physical activity decreased the odds. In light of pandemic-related restrictions associated with increased social isolation and psychological distress [49], people are more likely to smoke and engage in sedentary behaviors such as screen time which limits physical activity [50, 51] and increases the risk of metabolic diseases. A recent systematic review assessing screen-based sedentary behavior among adolescents during the COVID-19 pandemic reported a dose-response association between increased levels of screen time and components of metabolic syndrome [28]. Our findings are consistent with the existing literature on smoking as a risk factor for metabolic conditions [52–55] and increased physical activity as a protective factor [56–58]. As such, both smoking cessation and physical activity should be encouraged to reduce the risk of metabolic conditions, especially during this pandemic.

### Implications to practice and research

The COVID-19 pandemic has placed extraordinary demand on public health systems and essential services, while individuals with underlying health conditions such as diabetes, hypertension, and cardiovascular diseases are at higher risk of hospitalization and death [8, 59]. Thus, the association between COVID-19 and metabolic conditions alongside the disparities highlighted in this study suggests the need for further research and fair allocation of medical resources to address these conditions during and after the pandemic.

Although this study provides additional evidence to the literature on the effects of the COVID-19 pandemic related to metabolic conditions among the generally representative U.S. population, some limitations should be considered. These limitations include self-reporting bias and potential underestimation of chronic health problems, such as metabolic conditions, that develop over time. Given the cross-sectional data and lack of temporal sequence information on the variables, we could not make causal inferences. Longitudinal follow-up should be

continued for future research to further validate this study's findings. Additionally, public health assessment tools specifically validated for chronic diseases, such as metabolic conditions, that could be used for national observatory datasets would allow researchers to more rapidly evaluate data in real time in future public health crises, such as this recent global pandemic. Standardized validated tools would provide more meaningful assessments and results nationally and internationally. Moreover, there are probable effects of confounders not considered in this study such as sedentary behavior, sleep pattern, eating habits, and employment that might give rise to inaccurate estimates of the true association. Furthermore, the HINTS datasets do not contain responses specific to the COVID-19 pandemic. Additionally, because metabolic conditions and lifestyle behavioral risk factors take time to accumulate and change health conditions, the results of this study might be biased and under-estimated given the durations of the data before (January through April 2019) and during (February through June 2020) the COVID-19 pandemic. Therefore, future studies comparing the rates and prevalence of metabolic conditions before and during the pandemic are needed considering the longer of the data.

Despite these limitations, these data validate that high-risk groups, such as advanced age, should be targeted for interventions to protect against the negative effects of COVID-19. Another gap in the literature that could be addressed with future research is the health consequences of a public lockdown, which was the mitigation strategy for a global pandemic, compared to the consequences of the infectious disease itself. Development of tools designed to measure outcomes secondary to each of these distinctly different effects would benefit future research and resultant health policy. For example, it would be helpful to know if depressive symptoms were a direct effect of the disease, such as suffering from long-COVID, or from the social isolation secondary to the public lockdown. Overall, the HINTS dataset provides an efficient means to evaluate important public health questions in a rapidly evolving situation such as the COVID-19 pandemic.

## Conclusion

In this nationally representative sample of U.S. adults, the prevalence of metabolic conditions increased during the COVID-19 pandemic in certain subgroups of individuals. Specifically, there was an increased risk of metabolic outcomes associated with older age. Other groups with signals for increased risk included: non-Hispanic Black people, former smokers, individuals with poor health status, and mild anxiety. Thus, there is a need for proper rationing of resources to address these conditions during the pandemic.

## Author Contributions

**Conceptualization:** Hadii M. Mamudu, Trishita Paul, David W. Stewart.

**Data curation:** Hadii M. Mamudu, David Adzrago, Emmanuel O. Odame, David W. Stewart.

**Formal analysis:** David Adzrago.

**Investigation:** Emmanuel O. Odame, Oluwabunmi Dada, Valentine Nriagu, David W. Stewart, Jessica Adams.

**Methodology:** David Adzrago, Jessica Adams.

**Supervision:** Hadii M. Mamudu.

**Validation:** Hadii M. Mamudu, David W. Stewart.

**Visualization:** Hadii M. Mamudu.

**Writing – original draft:** David Adzrago, Emmanuel O. Odame, Oluwabunmi Dada, Trishita Paul.

**Writing – review & editing:** Hadii M. Mamudu, David Adzrago, Emmanuel O. Odame, Oluwabunmi Dada, Florence W. Weierbach, Karilynn Dowling-McClay, David W. Stewart, Timir K. Paul.

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
