## [Decision Letter · Decision Letter 0]

15 Sep 2022

PONE-D-22-21818The prevalence of cardiometabolic conditions before and during COVID-19 and its association with health and sociodemographic factorsPLOS ONE

Dear Dr. Mamudu,

Thank you for submitting your manuscript to PLOS ONE. After careful consideration, we feel that it has merit but does not fully meet PLOS ONE’s publication criteria as it currently stands. Therefore, we invite you to submit a revised version of the manuscript that addresses the points raised during the review process.

We look forward to receiving your revised manuscript.

Kind regards,

Taeyun Kim

Academic Editor

PLOS ONE

Journal Requirements:

2. Please amend the manuscript submission data (via Edit Submission) to include author Stewart DW,PharmD.

Reviewers' comments:

Reviewer's Responses to Questions

**Comments to the Author**

1. Is the manuscript technically sound, and do the data support the conclusions?

Reviewer #1: Partly

Reviewer #2: Partly

2. Has the statistical analysis been performed appropriately and rigorously? 

Reviewer #1: Yes

Reviewer #2: Yes

3. Have the authors made all data underlying the findings in their manuscript fully available?

Reviewer #1: No

Reviewer #2: Yes

4. Is the manuscript presented in an intelligible fashion and written in standard English?

Reviewer #1: Yes

Reviewer #2: No

5. Review Comments to the Author

Reviewer #1: This paper evaluates the population prevalence of diabetes, obesity and hypertension in the US during 2019 and 2020. The paper concludes that the prevalence of these conditions increased during the Covid-19 pandemic.

1. The title seems to imply that observations were made on Covid-19 status of individuals, which is not the case. Please adjust the wording of the title to make this clear.

2. In the Abstract, make clear that the study compares 2019 and 2020 and no observations were made on Covid-19 as such. We know that diabetes has been increasing over many years, so this paper does not demonstrate any deviation from the underlying trend, consquently the conclusions that can be darwn are quite limited.

3. Introduction, lines 68-69, there needs to be more discussion of references 16 and 21 - what do these studies show? Also, how widespread and what were the tiomings of lockdown orders in the target population for this study.

4. The outcome comprises hypertension, obesity and diabetes. No cardiac conditions are included, so this might be equated with the 'metabolic syndrome'.

5. In the analysis, explain what weights were employed.

6. Using a cut-off of P <0.05 is an out of date approach to interpretation. Please follow the ASA guidelines on P values. https://www.tandfonline.com/doi/full/10.1080/00031305.2016.1154108

7. Figure 1 may be better as a Table. If included it will be preferred to show the results for 2019 and 2020 side by side so these can be more readily compared.

8. Table 1. More appropriate column headers may be '2019' and '2020'.

9. Table 2: if the intention is to see whether associations differed in 2019 and 2020, it would be better to include all the data in a one model and test for the interaction of each variable with study year. Most of the associations appear to be quite similar across years.

10. In the Discussion, what can be concluded is very limited because the study has not evaluated secular trends over time. The difference between 2019 and 2020 could be accounted for by the underlying trend.

11. In the Limitations section, mention that no caulsa inferences can be drawn.

12. In the conclusion, where it says 'Disparities in cardiometabolic conditions became more evident after the pandemic', there does not appear to be sufficient evidence from the analysis to support this conclusion.

13. Where it says ' the prevalence of cardiometabolic conditions increased during the COVID-19 pandemic', only diabetes increased not the other conditions, and we do not know whether the increase exceeded pre pandemic expectations. Please address this text in teh Abstract also.

Reviewer #2: Review of Manuscript Number: PONE-D-22-21818

Introduction:

The introduction is lacking the part linking the risk factors especially mental health status during pandemic to cardiometabolic diseases and how COVID-19 pandemic has led to an increased adoption of such behavioral risk factors e.g., physical inactivity and smoking.

Line 62-64

COVID-19 may have an exacerbating effect on glycemic control for patients with diabetes [14, 20], and there may be risk of increased body mass index (BMI) as well as a deterioration in glucose regulation due to COVID-19 [16, 21].

Reviewer comments:

Will you clarify if COVID-19 infection or the implications of COVID-19 related lockdown have led to exacerbation of glycemic control and increased BMI?

Methods:

Line 90-91

Briefly, the 2019 survey (HINTS 5 Cycle 3) was conducted from January through April 2019, and the 2020 survey (HINTS 5 Cycle 4) was conducted from February through June 2020.

Reviewer comments:

COVID-19 was declared pandemic on March 11th, 2020, and lockdown has followed in most of the world regions. Therefore, if you are aiming to compare the rates and prevalence of cardiometabolic diseases, that are mainly linked to lifestyle behavioral risk factors taking time to accumulate and changing health conditions, before and during COVID-19, the results of this study might be biased and under-estimated.

Line 124-125

The number of days per week of moderate intensity physical activity (none and at least one day per week)

Reviewer comments:

Why have you chosen to ask none or at least one day per week? Physical activity significance should hit the international recommendations of 150 mins/week. Therefore, performing one time/week or so will not add significant information, hence correlation. If you are not using a valid tool to measure your variables, you will be subjected to bias. It is better to question about the days and minuets and calculate the mean.

Results

Line 147-151

prevalence of diabetes was higher during the COVID-19 pandemic (18.10%) than before the 150 pandemic (17.28%). However, the prevalence of hypertension (36.36% vs. 36.38%) and obesity 151 (34.68% vs. 34.18%) was similar during and before the pandemic.

Reviewer comments:

Again, the slight variations in the prevalence of cardiometabolic diseases between before and during the pandemic is most likely due to early trials on lifestyle related behavior that work in cumulative effects manner (developing over longer period) developing chronic diseases.

Line 163-166

Individuals who were former cigarette smokers or current smokers had increased prevalence of cardiometabolic conditions. For e cigarette use groups, the prevalence had increased for those who had never used e-cigarettes and those who currently used e-cigarettes however decreased for former e-cigarette users.

Reviewer comments:

The prevalence was increased for those who never smoked e-cigarettes and those who currently use e cigarette, is bringing so much confusion for the reader and later to decision makers. You need to revisit your analysis or at least explain your odd findings in the discussion section comparing to previous research.

Discussion

Line 200-202

Reviewer comments:

Discussion needs to be improved.

1. Correlation of age and risk of cardiometabolic conditions is not well highlighted. First, your study revealed that age group from 26- ≥65 year are at increased risk when compared to 18-25 years. I can notice that the risk is doubled for 26-34 year and tripled for 35-49 year. Therefore, in your recommendations, you need to focus not only on older age group but middle-aged as well.

2. You need to enrich discussion further to add new implications of your results. Since the findings are not adding additional knowledge to the literature, you need to discuss your odd findings and give explanations. You may discuss how to improve future research in the same area e.g., by adopting validated tools to measure variables, by using more reliable source of data like registry or objective measures as compared to self-reported. Suggestions in how to improve the internal validity of the results are important and show your understanding of pitfalls. Therefore, add a section of “implications to practice & research’.

Limitations:

Kindly add the limitations discussed above and the probable effect of confounders not considered in this study such as sedentary behavior, sleep, eating patterns, employment and etc., that might give rise to inaccurate estimates of the true association.

Overall, the manuscript needs improvement in English writing, linking ideas, relating results to previous findings, and most importantly providing explanation of each finding that disagree with the existing knowledge or literature.

Since the study design is cross-sectional, the main finding we are looking for is the prevalence of cardiometabolic conditions before and compare it to during the pandemic. However, your findings are not impressive and not reflecting the actual impact of COVID-19 on NCDs burden due to methodology reasons highlighted above. For this, you have to enrich your paper with additional values such as using the analysis of predictors and justifications of such findings.

Add the following article in your referencing (linking COVID-19 to behavioral risk factors which increase the risk of cardiometabolic conditions)

COVID-19 and screen-based sedentary behaviour: Systematic review of digital screen time and metabolic syndrome in adolescents | PLOS ONE

Available in PubMed also: COVID-19 and screen-based sedentary behaviour: Systematic review of digital screen time and metabolic syndrome in adolescents - PubMed (nih.gov)

6. PLOS authors have the option to publish the peer review history of their article (what does this mean?). If published, this will include your full peer review and any attached files.

Reviewer #1: **Yes: **Martin Gulliford

Reviewer #2: **Yes: **Sarah Rashid Musa

---

## [Author Response · Author response to Decision Letter 0]

16 Nov 2022

Response to Review of Manuscript Number: PONE-D-22-21818

Reviewer #1: This paper evaluates the population prevalence of diabetes, obesity and hypertension in the US during 2019 and 2020. The paper concludes that the prevalence of these conditions increased during the Covid-19 pandemic.

Comment

1. The title seems to imply that observations were made on Covid-19 status of individuals, which is not the case. Please adjust the wording of the title to make this clear.

Response 

We appreciate your comment. Our title implies the observations made before and during the COVID-19 pandemic:

“The prevalence of cardiometabolic conditions before and during COVID-19 and its association with health and sociodemographic factors.”

We have revised the title by adding “pandemic” to further clarify it as:

“The prevalence of metabolic conditions before and during the COVID-19 pandemic and its association with health and sociodemographic factors.”

Comment

2. In the Abstract, make clear that the study compares 2019 and 2020 and no observations were made on Covid-19 as such. We know that diabetes has been increasing over many years, so this paper does not demonstrate any deviation from the underlying trend, consequently the conclusions that can be drawn are quite limited.

Response 

We appreciate your comment. We have stated our study’s aim as:

“We examined the prevalence and association of cardiometabolic conditions with health and sociodemographic factors before and during the COVID-19 pandemic.”

We have revised the methods to further incorporate your suggestion as:

“Data were drawn from the 2019 (N= 5,359) and 2020 (N= 3,830) Health Information National Trends Surveys on adults to compare observations before (2019) and during (2020) the COVID-19 pandemic.”

Comment

3. Introduction, lines 68-69, there needs to be more discussion of references 16 and 21 - what do these studies show? Also, how widespread and what were the timings of lockdown orders in the target population for this study.

Response 

Thank you for this observation. We have revised the entire paragraph to incorporate your suggestion. For the lockdown timings, it varies for all studies in that paragraph and was summarized to prevent making that paragraph too long. Below is the explanation of the lockdown timings in the study. 

The timing of lockdown orders in reference 16 compared was before the lockdown and the 6th month of lockdown. The study population were a research cohort in the Istanbul Research and Education Hospital, Turkey, from March 2019 to October 2020.

In reference 20, the study population included all patients with diabetes mellitus who visited the Tohoku Medical and Pharmaceutical University Hospital in Sendai, from January 1, 2019, to August 31, 2020. Japan declared a state of emergency on April 7, 2020, so we presume this is the date their lockdown order started, although they think a state of emergency differs from lockdown orders established by other nationals.

In reference 21, the study population included outpatients at the Diabetology Unit of Humanitas Clinical and Research Center, IRCCS in Italy at baseline, between December 15, 2019, and March 1, 2020, and at the resumption of clinical activities, between May 15 and June 30, 2020.

In reference 22, the study population included Type 2 diabetes mellitus patients unable to attend clinic follow-up visits due to the lockdown order in Turkey between March 16 and June 1, 2020, but attended follow-ups in July and August 2020 after the restriction had been lifted.

Comment

4. The outcome comprises hypertension, obesity and diabetes. No cardiac conditions are included, so this might be equated with the 'metabolic syndrome'.

Response 

We agree with the reviewer; therefore, we have changed all “cardiometabolic” to “metabolic,” from the title to the conclusion. This is because we do not have data specifically on waist circumference, triglycerides, and HDL, therefore, using the term “metabolic syndrome” is not appropriate. As such, we will use the term “metabolic” instead of “cardiometabolic” as the reviewer suggested. 

Comment

5. In the analysis, explain what weights were employed.

Response

Thank you. We have included weight information as:

“The HINTS sampling weight was applied to the analysis to achieve population estimates and offset nonresponse.” (see the first sentence in the statistical analysis section)

Comment

6. Using a cut-off of P <0.05 is an out of date approach to interpretation. Please follow the ASA guidelines on P values. https://www.tandfonline.com/doi/full/10.1080/00031305.2016.1154108

Response

We appreciate your suggestion. We would like the reviewer to know that our decision on whether a result is statistically significant or not was informed by the p-value and the Confidence Interval (CI). As such, our decision-making approach is consistent with the recommendation by the American Statistical Association [https://www.tandfonline.com/doi/full/10.1080/00031305.2016.1154108]. Further, as recommended by Leo and Sardanelli [https://eurradiolexp.springeropen.com/articles/10.1186/s41747-020-0145-y], we have shown the actual p-value so that the reader could determine the extent of the association. Thus, our reporting of the results addresses the statistical issues raised by the reviewer.

Comment

7. Figure 1 may be better as a Table. If included it will be preferred to show the results for 2019 and 2020 side by side so these can be more readily compared.

Response

Thank you for the suggestion. We believe that a figure could better depict the patterns of the outcomes than a table and, as such, we would like to retain the figure. However, if the reviewer still feels that we should replace the figure with the table, we would be pleased to do.

Comment

8. Table 1. More appropriate column headers may be '2019' and '2020'.

Response 

Thank you. We have included the 2019 and 2020 in the column headers.

Comment

9. Table 2: if the intention is to see whether associations differed in 2019 and 2020, it would be better to include all the data in a one model and test for the interaction of each variable with study year. Most of the associations appear to be quite similar across years.

Response 

We appreciate your comment. Given the differences in the population in 2019 and 2020 based on the outcomes, we could not include all the data in one model. Respectfully, we would like to maintain the table in its current form to avoid any statistical falsification.

Comment

10. In the Discussion, what can be concluded is very limited because the study has not evaluated secular trends over time. The difference between 2019 and 2020 could be accounted for by the underlying trend.

Response 

We agree with you. As such, we have toned down on our interpretations and reasoning by not making causal inferences or making inferences beyond our results.

Comment

11. In the Limitations section, mention that no causal inferences can be drawn.

Response

Thank you. We have included your suggestion in the limitation section.

Comment

12. In the conclusion, where it says 'Disparities in cardiometabolic conditions became more evident after the pandemic', there does not appear to be sufficient evidence from the analysis to support this conclusion.

Response

Thank you for pointing this out. We have revised the statement as: 

“The prevalence of metabolic conditions increased during the COVID-19 pandemic in certain subgroups of individuals. Specifically, there was an increased risk of metabolic conditions associated with older age. Other groups with signals for increased risks include non -Hispanic Black people, former smokers, individuals with poor health status, and mild anxiety.”

Comment

13. Where it says ' the prevalence of cardiometabolic conditions increased during the COVID-19 pandemic', only diabetes increased not the other conditions, and we do not know whether the increase exceeded pre pandemic expectations. Please address this text in the Abstract also.

Response 

We have revised this in the abstract also as suggested: 

“This study found increased odds of metabolic conditions among certain subgroups of U.S. adults during the pandemic”

Reviewer #2:

Introduction:

The introduction is lacking the part linking the risk factors especially mental health status during pandemic to cardiometabolic diseases and how COVID-19 pandemic has led to an increased adoption of such behavioral risk factors e.g., physical inactivity and smoking. 

Line 62-64

COVID-19 may have an exacerbating effect on glycemic control for patients with diabetes [14, 20], and there may be risk of increased body mass index (BMI) as well as a deterioration in glucose regulation due to COVID-19 [16, 21].

Response: 

Thank you for your comment. This comment has been incorporated into the introduction section of the revised manuscript.

Reviewer comments:

Will you clarify if COVID-19 infection or the implications of COVID-19 related lockdown have led to exacerbation of glycemic control and increased BMI?

Response: 

Thank you for your comment. This statement has been clarified in the introduction of the revised manuscript. 

Methods:

Line 90-91

Briefly, the 2019 survey (HINTS 5 Cycle 3) was conducted from January through April 2019, and the 2020 survey (HINTS 5 Cycle 4) was conducted from February through June 2020.

Reviewer comments:

COVID-19 was declared pandemic on March 11th, 2020, and lockdown has followed in most of the world regions. Therefore, if you are aiming to compare the rates and prevalence of cardiometabolic diseases, which are mainly linked to lifestyle behavioral risk factors taking time to accumulate and changing health conditions, before and during COVID-19, the results of this study might be biased and under-estimated. 

Response:

Thank you for pointing out this information. We agree with the reviewer and have added the recommendation as a limitation of the study (see the limitation section).

Line 124-125

The number of days per week of moderate intensity physical activity (none and at least one day per week)

Reviewer comments:

Why have you chosen to ask none or at least one day per week? Physical activity significance should hit the international recommendations of 150 mins/week. Therefore, performing one time/week or so will not add significant information, hence correlation. If you are not using a valid tool to measure your variables, you will be subjected to bias. It is better to question about the days and minutes and calculate the mean.

Response

We appreciate your comments. This variable is a standardized measure in HINTS and based on the number of days per week of moderate intensity physical activity. The “150 mins/week” is used to derive the “moderate physical activity” either per day or week, hence the choice for this study.

Results

Line 147-151

prevalence of diabetes was higher during the COVID-19 pandemic (18.10%) than before the 150 pandemic (17.28%). However, the prevalence of hypertension (36.36% vs. 36.38%) and obesity 151 (34.68% vs. 34.18%) was similar during and before the pandemic.

Reviewer comments:

Again, the slight variations in the prevalence of cardiometabolic diseases between before and during the pandemic is most likely due to early trials on lifestyle related behavior that work in cumulative effects manner (developing over longer period) developing chronic diseases.

Response 

Thank you for noting this. This suggestion has been included in the limitations of the revised manuscript (see the limitation section).

Line 163-166

Individuals who were former cigarette smokers or current smokers had increased prevalence of cardiometabolic conditions. For e-cigarette use groups, the prevalence had increased for those who had never used e-cigarettes and those who currently used e-cigarettes however decreased for former e-cigarette users. The prevalence was increased for those who never smoked e-cigarettes and those who currently use e-cigarette, is bringing so much confusion for the reader and later to decision makers. You need to revisit your analysis or at least explain your odd findings in the discussion section comparing to previous research. 

Response

Thank you! We revisited the analysis and found the same results. This finding is consistent with the tobacco literature: former tobacco users, including former e-cigarette users, are less likely to engage in unhealthy behaviors; therefore, they are less likely to develop health conditions such as cardiometabolic conditions (https://nida.nih.gov/publications/research-reports/tobacco-nicotine-e-cigarettes/what-are-physical-health-consequences-tobacco-use;
https://www.health.ny.gov/prevention/tobacco_control/; Lowe et al., 2009; U.S. Department of Health and Human Services, 2014). However, the pandemic has altered many lifestyles, including tobacco use behaviors that might have also affected their risks for cardiometabolic conditions.

Citations:

Lowe, F. J., Gregg, E. O., & McEwan, M. (2009). Evaluation of biomarkers of exposure and potential harm in smokers, former smokers, and never-smokers. Clinical chemistry and laboratory medicine, 47(3), 311-320.

U.S. Department of Health and Human Services (2014). The Health Consequences of Smoking—50 Years of Progress: A Report of the Surgeon General. Atlanta, GA: U.S. Department of Health and Human Services, Centers for Disease Control and Prevention, National Center for Chronic Disease Prevention and Health Promotion, Office on Smoking and Health. https://www.hhs.gov/sites/default/files/consequences-smoking-exec-summary.pdf

Discussion

Line 200-202

Reviewer comments:

Discussion needs to be improved. 

1. Correlation of age and risk of cardiometabolic conditions is not well highlighted. First, your study revealed that age group from 26- ≥65 year are at increased risk when compared to 18-25 years. I can notice that the risk is doubled for 26-34 year and tripled for 35-49 year. Therefore, in your recommendations, you need to focus not only on older age group but middle-aged as well. 

Response 

Thank you! We have included discussions on these middle-aged groups as suggested in the revised manuscript. For instance, we found that the odds of cardiometabolic outcomes were significantly higher only among the elderly age groups (50-64, and 65+) compared to young adults before the pandemic. However, these increases in odds almost doubled among these age groups during the pandemic. Significantly higher odds were also noted in the middle age groups (26-34 and 35-49), where the odds almost tripled for the 35-49 age group during the pandemic.

Comment:

2. You need to enrich discussion further to add new implications of your results. Since the findings are not adding additional knowledge to the literature, you need to discuss your odd findings and give explanations. You may discuss how to improve future research in the same area e.g., by adopting validated tools to measure variables, by using more reliable source of data like registry or objective measures as compared to self-reported. Suggestions in how to improve the internal validity of the results are important and show your understanding of pitfalls. Therefore, add a section of “implications to practice & research.”

Response 

Thank you for this suggestion. We have added a new section of “Implications to Practice and Research,” including limitations of the study, in the discussion section. 

Regarding the issue of validity and reliability, as in most national surveys, the measures in HINTS have been validated and widely used since 2002/2003. We have, however, noted the limitation of using self-reported measures and recommended using objective measures in future studies.

Regarding the internal validity of the results, we followed the analytical recommendations of HINTS in computing accurate estimates. As such, we believe our estimates are accurate and valid. 

Limitations:

Comment

Kindly add the limitations discussed above and the probable effect of confounders not considered in this study such as sedentary behavior, sleep, eating patterns, employment and etc., that might give rise to inaccurate estimates of the true association.

Response

Thank you. We have incorporated your suggestions in the revised manuscript (see the limitation section).

Comment

Overall, the manuscript needs improvement in English writing, linking ideas, relating results to previous findings, and most importantly providing explanation of each finding that disagree with the existing knowledge or literature.

Response

We have thoroughly reviewed the entire paper by doing line-by-line editing. Additionally, we asked our colleague not familiar with the study to review the paper for language.

Comment

Since the study design is cross-sectional, the main finding we are looking for is the prevalence of cardiometabolic conditions before and compare it to during the pandemic. However, your findings are not impressive and not reflecting the actual impact of COVID-19 on NCDs burden due to methodology reasons highlighted above. For this, you have to enrich your paper with additional values such as using the analysis of predictors and justifications of such findings.

Response 

We agree with the reviewer that the cross-sectional data is a limitation of this study. As such, we have included this limitation in the limitations section of the revised manuscript. In future studies, we will consider longitudinal data for the analysis of predictors and justifications of such findings. Additionally, we have enhanced the discussion by relating our study to the extant literature and illuminating its added value to the growing literature on COVID-19.

Comment

Add the following article in your referencing (linking COVID-19 to behavioral risk factors which increase the risk of cardiometabolic conditions)

COVID-19 and screen-based sedentary behaviour: Systematic review of digital screen time and metabolic syndrome in adolescents | PLOS ONE

Available in PubMed also: COVID-19 and screen-based sedentary behaviour: Systematic review of digital screen time and metabolic syndrome in adolescents - PubMed (nih.gov)

Response 

The article has been incorporated in the revised manuscript. Specifically, we have incorporated the following in the discussion section: 

“Sedentary behaviors such as screen time which limits physical activity [50, 51], increases the risk of cardiometabolic diseases. A recent systematic review assessing screen-based sedentary behavior among adolescents during the COVID-19 pandemic reported a dose-response association between increased level of screen time and components of metabolic syndrome [Musa et al]”

---

## [Decision Letter · Decision Letter 1]

28 Nov 2022

PONE-D-22-21818R1The prevalence of metabolic conditions before and during the COVID-19 pandemic and its association with health and sociodemographic factorsPLOS ONE

Dear Dr. Mamudu,

Thank you for submitting your manuscript to PLOS ONE. After careful consideration, we feel that it has merit but does not fully meet PLOS ONE’s publication criteria as it currently stands. Therefore, we invite you to submit a revised version of the manuscript that addresses the points raised during the review process.

We look forward to receiving your revised manuscript.

Kind regards,

Taeyun Kim

Academic Editor

PLOS ONE

Journal Requirements:

Reviewers' comments:

Reviewer's Responses to Questions

**Comments to the Author**

1. If the authors have adequately addressed your comments raised in a previous round of review and you feel that this manuscript is now acceptable for publication, you may indicate that here to bypass the “Comments to the Author” section, enter your conflict of interest statement in the “Confidential to Editor” section, and submit your "Accept" recommendation.

Reviewer #1: All comments have been addressed

Reviewer #2: All comments have been addressed

2. Is the manuscript technically sound, and do the data support the conclusions?

Reviewer #1: Yes

Reviewer #2: Yes

3. Has the statistical analysis been performed appropriately and rigorously? 

Reviewer #1: Yes

Reviewer #2: Yes

4. Have the authors made all data underlying the findings in their manuscript fully available?

Reviewer #1: Yes

Reviewer #2: Yes

5. Is the manuscript presented in an intelligible fashion and written in standard English?

Reviewer #1: Yes

Reviewer #2: Yes

6. Review Comments to the Author

Reviewer #1: 1. In the Abstract, suggest changing 'rationing' to 'allocation'.

2. In the Abstract, where it refers to ' older adults, non-Hispanic Black people... individuals with poor health status', no data are presented to support this statement in the Abstract. Eitehr include supporting evidence in the Abstract or omit.

Reviewer #2: Thank you for addressing every point and incorporating them into your revised manuscript.

The paper sounds much better indeed with feedback of reviewer 1 as well.

7. PLOS authors have the option to publish the peer review history of their article (what does this mean?). If published, this will include your full peer review and any attached files.

Reviewer #1: No

Reviewer #2: **Yes: **Dr. Sarah Musa

---

## [Author Response · Author response to Decision Letter 1]

30 Nov 2022

Reviewer #1: 

Comment

1. In the Abstract, suggest changing 'rationing' to 'allocation'.

Response

We have changed it as suggested.

Comment

2. In the Abstract, where it refers to ' older adults, non-Hispanic Black people... individuals with poor health status', no data are presented to support this statement in the Abstract. Either include supporting evidence in the Abstract or omit.

Response 

We have deleted the emphasis as suggested.

Reviewer #2: 

Comment

Thank you for addressing every point and incorporating them into your revised manuscript.

The paper sounds much better indeed with feedback of reviewer 1 as well.

Response

Thank you for helping to improve the paper substantively and stylistically.

---

## [Editor Report · Decision Letter 2]

7 Dec 2022

The prevalence of metabolic conditions before and during the COVID-19 pandemic and its association with health and sociodemographic factors

PONE-D-22-21818R2

Dear Dr. Mamudu,

We’re pleased to inform you that your manuscript has been judged scientifically suitable for publication and will be formally accepted for publication once it meets all outstanding technical requirements.

Kind regards,

Taeyun Kim

Academic Editor

PLOS ONE
---

## [Editor Report · Acceptance letter]

2 Feb 2023

PONE-D-22-21818R2 

The prevalence of metabolic conditions before and during the COVID-19 pandemic and its association with health and sociodemographic factors 

Dear Dr. Mamudu:

I'm pleased to inform you that your manuscript has been deemed suitable for publication in PLOS ONE. Congratulations! Your manuscript is now with our production department. 

Kind regards, 

on behalf of

Dr. Taeyun Kim 

Academic Editor

PLOS ONE